# Revisiting a Distinct Entity in Pulmonary Vascular Disease: Chronic Thromboembolic Pulmonary Hypertension (CTEPH)

**DOI:** 10.3390/medicina57040355

**Published:** 2021-04-07

**Authors:** Munish Sharma, Deborah Jo Levine

**Affiliations:** 1Corpus Christi Medical Center, Department of Pulmonary Medicine, Corpus Christi, TX 78412, USA; munishintmed@gmail.com; 2Department of Pulmonary Medicine, University of Texas Health Science Center, San Antonio, TX 78229, USA

**Keywords:** chronic thromboembolic pulmonary hypertension, pulmonary embolism, pulmonary thromboendarterectomy, percutaneous balloon pulmonary angioplasty, riociguat, ventilation-perfusion lung scan, computed tomography pulmonary angiography, pulmonary angiography

## Abstract

Chronic thromboembolic pulmonary hypertension (CTEPH) is a specific type of pulmonary hypertension (PH) and the major component of Group 4 pulmonary hypertension (PH). It is caused by pulmonary vasculature obstruction that leads to a progressive increase in pulmonary vascular resistance and, ultimately, to failure of the right ventricle. Pulmonary thromboendarterectomy (PEA) is the only definitive therapy, so a timely diagnosis and early referral to a specialized PEA center to determine candidacy is prudent for a favorable outcome. Percutaneous balloon pulmonary angioplasty (BPA) has a potential role in patients unsuitable for PEA. Medical therapy with riociguat is the only PH-specific medical therapy currently approved for the treatment of inoperable or persistent CTEPH. This review article aims to revisit CTEPH succinctly with a review of prevailing literature.

## 1. Introduction

The 6th world symposium on pulmonary hypertension (PH) in 2018 has defined PH as an elevated mean pulmonary artery pressure (mPAP) ≥ 20 mmHg at rest, pulmonary capillary wedge pressure < 15 mmHg, and pulmonary vascular resistance (PVR) ≥ 3 Wood Unit (WU) [1]. PH can be further subdivided into 5 groups (Table 1). 

Chronic thromboembolic pulmonary hypertension (CTEPH) belongs to Group 4 of the PH classification. CTEPH is defined as an obstruction of the pulmonary arterial vasculature secondary to single or recurrent episodes of pulmonary embolism (PE) [2]. It is a distinct and infrequently diagnosed entity of PH that is progressive and can be fatal if left untreated.

## 2. Epidemiology

CTEPH occurs in a minority of the patients who fail to restore normal pulmonary perfusion after single or recurrent episodes of PE. It generally develops within the first year after PE and is an uncommon occurrence after about 2 years of an episode of PE [2]. The exact incidence of CTEPH is difficult to determine but it is estimated to be prevalent in around 0.5–5% of patients that have an acute PE [3,4,5,6]. It has been reported in as many as 11.6% of patients with a history of recurrent PE [7] (Table 2).

Determining an accurate incidence of CTEPH after PE is challenging because of inconsistencies in reporting. An international registry for CTEPH from North America and Europe reported that up to 75% of patients with CTEPH had a definite proceeding episode of PE [8]. In contrast, data from Japan reported antecedent PE in around 15–33% of patients with CTEPH [9,10]. Studies from Japan have shown 80% female preponderance, while those from North America and Europe have reported around 49.9% incidence in females [8,9]. The nonspecific presentation of the disease, underutilization of the guideline-recommended ventilation-perfusion (V/Q) scan for screening, and lack of expertise in interpreting nuclear and radiological studies have made it difficult to quantify the burden of CTEPH more uniformly across the world. 

## 3. Pathophysiology of CTEPH

The current understanding of the pathophysiology of CTEPH indicates a more complex phenomenon than mere chronic obstruction of pulmonary vascular bed by unresolved thrombus and subsequent right ventricular (RV) dysfunction. Chronic obstruction of proximal pulmonary vessels by fibrotic clots, development of small vessel disease, and progressive diffuse remodeling of the pulmonary vasculature collectively represent the pathophysiology of CTEPH [10,11]. 

Nonresolution of thrombus after PE occurs only in a minority of patients. This leads to failure in the restoration of normal hemodynamics in the pulmonary circulatory system. In acute PE, fresh red clots made of red blood cells and fibrin mesh are easily detachable from the wall of the pulmonary vessels. In contrast, chronic clots are yellow-colored and composed of abundant inflammatory cells, elastin, and collagen fibers [10,12]. Further organization of the chronic clot impairs blood flow and leads to CTEPH development. This organization and fibrosis represent the characteristic bands and webs in a pulmonary angiogram [13]. Simonneau G. et al. [10] and Lang IM et al. [14] have summarized the pathophysiology of CTEPH as depicted in Figure 1. 

## 4. Predisposing Factors for CTEPH

Various characteristics have been implicated with the development of CTEPH after a PE (Table 3). The most commonly proposed risk factors are large persistent or recurrent pulmonary emboli, and insufficient anticoagulation [10,14]. Malignancy has also been suspected to be a risk factor for the development of CTEPH by limited studies. It is likely related to the increased production of cytokines, acute phase reactants, and unregulated coagulation and fibrinolytic system. In a European study that involved 687 patients, there was a strong association between the development of CTEPH in patients with malignancy, with an odds ratio of 3.76 and 95% confidence in the wall of 1.47–10.43 [15].

Chronic infection has also been proposed as a predisposing factor for CTEPH in some studies. In a study involving patients with CTEPH and ventriculo-atrial shunts, DNA of staphylococcus species was isolated in six out of seven cases of pulmonary end-arterectomy [16].

There appears to be an association between CTEPH and a proinflammatory state. Elevated levels of interleukins 6 and 8 and interferon-gamma-induced protein-10 were found in a prospective study of patients with CTEPH [17]. Another study revealed that there was a significant reduction in elevated levels of C-reactive protein after pulmonary endarterectomy in 79 patients with CTEPH as compared to those without CTEPH [18]. 

Biological abnormalities have also been suspected to give rise to nonresolution of thrombus in patients with PE. Levels of some of the hereditary prothrombotic risk factors such as protein C, protein S, and deficiency of antithrombin levels are no different in CTEPH patients compared to the healthy patient population [19]. However, some prospective studies have shown increased incidences of antiphospholipid antibody, lupus anticoagulant, and clotting factors VIII and V on Willebrand factor in patients with CTEPH as compared to normal and PAH patients [20]. Studies have implicated more than 1600 genes in the pathogenesis of CTEPH. Studies have used oligonucleotide microarrays to investigate gene expression in endothelial cells of pulmonary arteries [21]. Upregulation or downregulation of the genes such as those responsible for the expression of interleukin-8, bone morphogenetic protein type 2 receptor (BMPR2), and polymorphism of angiotensin-converting enzyme gene have been described as contributing factors for CTEPH [21,22].

CTEPH is more common in patients with non-O blood groups. Around 77% of patients with CTEPH have blood group A, B, or AB as compared to 58% in PAH [20]. CTEPH patients also tend to have an abnormal level of fibrinogen molecules in the blood, which can lead to increased fibrin clots by the cross-linking of alpha chains [23]. Thyroid replacement therapy and splenectomy are also thought to be the risk factors by underlying platelet activation. There are higher platelet turnover and dysfunction in patients with CTEPH in comparison to normal controls [24]. Though multiple other risk factors have also been investigated, the development of CTEPH is most likely through a complex interaction of a multitude of factors rather than a single determinant.

## 5. Clinical Features

Progressive dyspnea and exercise intolerance are two main clinical features of CTEPH that lead patients to seek medical attention. These symptoms are nonspecific to CTEPH. Additional features of RV dysfunction such as pedal edema, exertional chest pain, extreme fatigue, syncope, and presyncope are seen later during the disease [25]. Early identification of this disease relies on a detailed patient history. Exercise intolerance and dyspnea on exertion are important symptoms to identify. It is imperative to inquire about the history of acute venous thromboembolism (VTE) or features suggestive of it such as lower extremity swelling, pain or discomfort, cough, dyspnea, and pleuritic chest pain. It has been found that as many as 38% of patients with CTEPH may not have a clear history of preceding VTE [26]. Thus, in the context of unexplained dyspnea, physicians should not be discouraged by the absence of clear antecedent VTE events and should still pursue CTEPH as a possible underlying cause.

Physical examination findings in CTEPH may vary based on the course of the disease. Subtle accentuation and narrowing in the splitting of the second heart sound (P2-pulmonic component) may be found in the early phase. Jugular venous distension with prominent ‘a’ and ‘v’ waves, right parasternal heave, fixed splitting of second heart sound, third heart sound (S3), pan systolic murmur of tricuspid regurgitation best heard at left lower sternal border and prominent with inspiration, early diastolic decrescendo murmur of pulmonary insufficiency, pulmonary artery bruits heard over the lungs, pitting pedal edema, ascites, and hepatomegaly are features of RV dysfunction that can be appreciated later in the disease course [27]. 

## 6. Diagnostic Evaluation in CTEPH

Early diagnosis is of paramount importance in CTEPH so that timely initiation of therapeutic options can be undertaken. For a definite diagnosis of CTEPH, the aforementioned hemodynamic parameters for PH should be satisfied [1], along with evidence of proximal or distal thromboembolic occlusion of the pulmonary vasculature [27]. It should be noted that d-dimer has been found to be a nonspecific test and cannot be used to rule out CTEPH despite its high negative predictive value [28]. Since the initial symptoms of CTEPH, such as progressive dyspnea and exercise intolerance, are nonspecific and represent a variety of other pathologies, screening tests are useful in suspected cases. It is important to rule out common differential diagnoses such as obstructive or restrictive ventilatory defects, cardiomyopathy, coronary artery disease, and deconditioning. 

Echocardiography is a useful noninvasive means of screening patients at risk for developing CTEPH. It is not a definitive diagnostic tool and can be more useful for screening symptomatic patients with relevant risk factors. Doppler evaluation of maximum tricuspid regurgitation velocity (TRVmax) added to the right atrial pressure (RAP) is a useful tool in estimating pulmonary artery systolic pressure (PASP). TRVmax > 3.4 m/s is considered to represent high probability of PH in general, while TRVmax < 2.8 m/s represents low probability [29]. A significant number of patients with severe PH may not have an accurate TR signal on doppler and TRVmax might grossly underestimate the severity of PASP [30]. Right atrial enlargement, right ventricular systolic function, and leftward displacement of the interventricular septum are some other parameters that can be utilized in patients with CTEPH [27]. Pulmonary function test (PFT) is used to rule out obstructive and restrictive defects, and an isolated out-of-proportion decrease in diffusing capacity for carbon monoxide (DLCO) hints toward probable underlying pulmonary vascular defect. Arterial blood gas may show an increased A-a gradient with the low partial pressure of oxygen, especially with exertion [27]. Chest radiography may show differential areas of hypo-perfusion and hyper-perfusion, enlargement of main pulmonary arteries, and enlargement of the right atrium or right ventricle [27]. 

### 6.1. Ventilation-Perfusion Lung Scan, Computed Tomography Pulmonary Angiography, and Magnetic Resonance Angiography in CTEPH

In patients with stronger suspicion of CTEPH, more definitive diagnostic tools should be utilized to confirm the diagnosis. Although ventilation-perfusion lung scanning (V/Q) is not the gold-standard study for the diagnosis of CTEPH, it is a good screening tool. V/Q scan is preferred over computed tomography pulmonary angiography (CTPA) as the initial imaging of choice mainly due to its greater sensitivity. Tunirau N. et al. conducted a retrospective analysis of 227 patients referred to a pulmonary hypertension center in London, UK, between 2000 and 2005. They retrospectively compared the results of the V/Q scan and CTPA. V/Q scan was found to have a sensitivity ranging from 96–97.4% and specificity of 90–95%, while CTPA showed 51% sensitivity and 99% specificity [30]. In a more recent prospective study of 114 consecutive patients with CTEPH conducted by He J. et al. in 2012, the authors found a sensitivity of V/Q of 100%, which was quantitatively better than a CTA of 92.2%. However, the authors concluded that the results were similar and that both V/Q scan and CTA had excellent efficacy [25,31]. This might be due to advancements in imaging techniques and more accurate interpretation skills. Regardless, the V/Q scan is still favored by the majority of clinicians due to its utility in differentiating proximal larger pulmonary vessel involvement in CTEPH from more distal small vessel pathology in Group 1 PH [27]. Single or multiple larger mismatches found on the V/Q suggests CTEPH in contrast to smaller subsegmental defects seen in Group 1 PH [27]. 

The V/Q scan has its limitations in terms of specificity for CTEPH. Larger segmental defects may represent other pulmonary obstructive disorders such as pulmonary veno-occlusive disease (PVOD), sarcoma of the pulmonary artery, extrinsic compression of the pulmonary vasculature, and fibrosing mediastinitis [27]. The V/Q scan is often believed to underestimate the severity of pulmonary artery occlusion. In a study performed in 1988 by Ryan KL et al., 25 patients with CTEPH who had initial V/Q scans and subsequently underwent thromboendarterectomy were analyzed. The authors found no reliable correlation between the severity of pulmonary vascular obstruction and defects on V/Q scan when the patients ultimately underwent surgery [32].

Only a few studies have attempted to assess the role of magnetic resonance imaging (MRI) and magnetic resonance angiography (MRA) in the diagnostic evaluation of CTEPH. Rajaram S. et al. evaluated 132 consecutive patients referred to a pulmonary hypertension center in the United Kingdom undergoing lung perfusion MRI, V/Q scan, CTPA, and RHC [33]. Of the 132 patients, 78 were diagnosed with CTEPH. Lung perfusion MRI correctly diagnosed 76 out of 78 patients with CTEPH with an overall sensitivity of 96%, which was comparable to 96% sensitivity of V/Q scan and 94% sensitivity of CTPA [33]. MRI and MRA might have an added advantage, as an ionizing radiation tool is not utilized. More studies would be needed to better establish the role of MRI/MRA in CTEPH.

### 6.2. Role of Right Heart Catheterization and Pulmonary Angiography

If the V/Q scan is suggestive of CTEPH, additional testing is required with right heart catheterization (RHC) and pulmonary angiography. RHC should be performed not only to confirm the PH but also to assess the severity of the disease by obtaining complete hemodynamic parameters. In some cases, it might be beneficial to record hemodynamics after a brief period of exercise. Since the ability of the pulmonary artery to distend in response to an increase in cardiac output is lost in CTEPH, a disproportionate increase in pulmonary artery pressure can be seen [27]. 

A pulmonary angiogram is considered the gold-standard diagnostic test for CTEPH. It is used to assess if the thrombotic occlusions are surgically amenable and to confirm that the elevated PVR is not from secondary arteriopathy, distal vessel disease, or pulmonary venous hypertension [27]. Concern regarding the deterioration of hemodynamics after direct bolus administration of nonionic contrast and oxygen inhalation during pulmonary angiography has been voiced frequently. Pitton MB et al. performed a study in 1996 to address this concern [34]. Hemodynamic parameters were measured during pulmonary angiography in 33 patients who received oxygen and bolus injection of nonionic contrast. Comparison between a control group (11 patients), moderate PH group (9 patients), and severe PH group (13 patients) showed that oxygen contributed to greater safety. After pulmonary angiography PA pressure increased moderately, PVR and cardiac index remained unchanged, and systemic vascular resistance decreased slightly in the severe PH group. The overall systemic blood pressure decreased but was not found to pose significant safety issues [34]. Major angiographic patterns seen in CTEPH are pouch defects, webs or bands of the pulmonary artery, irregularities of the intima, abrupt pulmonary artery narrowing, and obstruction of lobar and segmental vessels [27].

## 7. Management of CTEPH

Current management strategies in CTEPH are based on the guidelines established by the 6th World Symposium on PH [1,2]. Current treatment modalities are outlined below (Figure 2). Lifelong anticoagulation is recommended for all patients with CTEPH [1,2]. The goal of long-term anticoagulation is to prevent not only the recurrence of VTE but also in situ thrombosis of the pulmonary artery [2,35]. Optimal anticoagulation can be initiated with parenteral agents such as unfractionated heparin or low molecular weight heparin before transitioning to warfarin with a goal international normalized ratio. It should be acknowledged that the use of long-term anticoagulation in CTEPH is based on the extrapolation of data from studies on single or recurrent PE, clinical experience, and expert recommendations.

### 7.1. Pulmonary Thromboendarterectomy (PEA) in CTEPH

PEA is the surgical treatment of choice. All patients with CTEPH should be evaluated for PEA regardless of symptomatology or abnormalities in hemodynamic parameters. Early referral to a center equipped with resources for PEA is key in the management of CTEPH. The decision to pursue PEA in CTEPH is dependent upon four criteria: Accessibility of thrombi, impairment in hemodynamic parameters or ventilatory compromise, comorbidities affecting peri- and postoperative outcomes, and patient acceptance [2,36].

Thrombi situated in the proximal pulmonary vasculature and the main, lobar, and segmental arteries are more easily amenable to PEA. The location and extent of thrombi determined by pulmonary angiogram are vital in determining the benefit of PEA [2]. If the proximal thromboembolic burden makes up the majority of the total thromboembolic, then only PEA can be expected to lower PVR post-surgery [37]. Hemodynamic parameters obtained from RHC are important in the final decision of whether to proceed with PEA. Severely elevated PVR at rest (>600–1200 dynes s/cm^5^; 7.5 to 15 Wood Units) are evident in most cases, but even in patients with PVR > 300 dynes s/cm^5^; 3.75 Wood Units, PEA can still be contemplated if progression of the disease is a concern [36]. If the hemodynamic worsens with exercise, these patients are at risk of further faster progression of the disease and can thus be considered for surgery [2,36]. Occasionally, patients with single main pulmonary artery obstruction can have exercise intolerance, mainly due to the increased dead space ventilation and higher requirement for minute ventilation without substantial derangement in hemodynamics. Such patients are also deemed to be potential candidates for PEA [2,38]. Perioperative and short- and long-term postoperative outcomes might depend on comorbid conditions. These should be carefully evaluated. The patient and the family members have to be involved in extensive discussion about the potential risks versus benefits of PEA. Ultimately, patient acceptance is a must to proceed with PEA (Figure 3). 

As per the international registry of CTEPH, the 3-year survival of patients undergoing PEA is 90% as compared to 70% in those without surgery [38]. In another study where a larger cohort was followed for 10 years, survival was 72% [25,39] (Figure 4). 

### 7.2. Percutaneous Pulmonary Balloon Angioplasty (BPA) in CTEPH

BPA is mainly used in patients with inoperable CTEPH (i.e., comorbidities not deemed suitable for PEA, residual PH after PEA) [40,41]. A study involving 500 consecutive patients with CTEPH, of which 97 underwent BPH, showed that the success rate was higher and complications were lower in ring-like stenosis as well as web lesions [41]. Mizoguchi H. et al. performed refined BPA in 68 consecutive patients with inoperable CTEPH [42]. They used intravascular ultrasound to determine the appropriate balloon size and performed BPA in a staged fashion, with a total of four sessions per patient with three vessels dilatation in each session. They were able to show an improvement in the World Health Organization (WHO) functional class for PH from III to II, with a *p*-value of <0.01 and a decrease in mPAP from 45.4 ± 9.6 mmHg before the BPA to 24 ± 6.4 mmHg. There was one death 28 days after BPA due to right ventricular failure. In the study, 41 patients developed reperfusion injury, 4 of which required mechanical ventilation [42]. Data from a multicenter registry of a total of 308 patients from 7 different institutions in Japan showed that mPAP improved from 43.2 ± 11.0 to 24.3 ± 6.4 mmHg after BPA. The complication rate was 36.3%, which included pulmonary injury in 17.8%, hemoptysis in 14.0%, and perforation of the pulmonary artery in 2.9% of patients. Of the 308 patients, 12 (4%) died during follow-up, with 8 deaths within 30 days post-BPA. Major complications associated with BPA are reperfusion injury, perforation of the pulmonary artery, dissection of the pulmonary artery, and hemoptysis [2]. 

### 7.3. PH-Targeted Medical Therapy in CTEPH

PH-targeted medical therapy is another approach to managing CTEPH in patients who are not operative candidates. Around 40% of CTEPH patients are considered inoperable and may potentially benefit from PH-targeted medical therapy [25]. Riociguat, a soluble guanylate cyclase agent, is the only approved medical therapy for inoperable CTEPH in many countries. The approval of riociguat was based on the results of multicenter, randomized, double-blinded CHEST-1 and CHEST-2 trials [43,44]. The CHEST-1 trial revealed a decrease in PVR by 226 dynes s/cm^5^ in the riociguat group as compared to an increase in PVR by 23 dynes s/cm^5^ in the placebo group (*p* < 0.001). An increase in 6-minute walk distance (6MWD) by a mean of 39 m in the riociguat group was also found, in contrast to a decrease by 6 m in the placebo group (*p* < 0.001). The riociguat group was also associated with an improvement in N-terminal pro-brain natriuretic peptide (NT-pro BNP) level and WHO functional class [43]. CHEST-2 was an open-label extension study in which eligible patients from CHEST-1 were entered. All these patients were given riociguat at an individualized dose up to a maximum dose of 2.5 mg thrice daily [44]. Overall survival was 93% (95% confidence interval of 89% to 96%) at 2 years. Improvement in 66MWD and NT-pro BNP were found to be important prognostic factors [44]. The MERIT-1 trial evaluated the use of macitentan, an endothelin receptor antagonist, in the treatment of inoperable CTEPH [45]. This trial also assessed the possibility of combination therapy in CTEPH as 61% of patients were already under treatment with phosphodiesterase type 5 inhibitors with or without oral or inhaled prostanoids [25,45]. Results of MERIT-1 revealed an improvement in 6MWD, PVR, and NT-pro BNP among other indices, with *p* values of 0.033, 0.041, and 0.040, respectively [45]. In 2008, BENEFiT (Bosentan Effects in iNopErable Forms of chronIc Thromboembolic pulmonary hypertension), a double-blinded, randomized, placebo-controlled trial, aimed to determine the effect of Bosentan (endothelin receptor antagonist) on pulmonary hemodynamics and exercise capacity in patients with inoperable CTEPH or recurrent/residual CTEPH after PEA [46]. BENEFiT included 157 patients (80 placebo group/77 treatment group). Whereas there was a statistically significant improvement in PVR and cardiac index, there was no significant improvement in 6MWD with bosentan [46]. The use of PH-specific medical therapy preoperatively is not routinely indicated but it is being increasingly used. A retrospective analysis of 11 patients treated with PH-specific medical therapy compared to the 244-control group showed minimal to no effect on pre-PEA hemodynamics and post-PEA outcomes [47]. 

Bilateral lung transplantation may be a surgical treatment option for patients who are not a candidate for or who have failed PEA, BPH, or PH-targeted medical therapy. Patients with CTEPH must meet the standard guideline for lung transplantation. More importantly, more data are required in patients who have undergone lung transplantation for CTEPH for future guidance and direction [2,36]. 

## 8. Conclusions

CTEPH is an infrequently diagnosed clinical entity and can be regarded as a distinct subtype within the entire spectrum of PH. It is progressive and can have devastating patient outcomes if not diagnosed and managed early. The aim of prevailing surgical, medical, and interventional modalities of treatment is to restore normal hemodynamics of pulmonary vasculature promptly. Failure to do so invariably inflicts irreparable damage as right ventricular failure ensues. For an early and accurate diagnosis of CTEPH and optimal utilization of treatment options, a high degree of clinical suspicion based on symptomatology and surveillance studies are required in patients with a recent or recurrent history of PE. There also has to be a growing awareness about CTEPH per se among clinicians and patients at risk. More specialized centers should be established with a multidisciplinary team comprising of medical, surgical, and interventional physicians with relevant expertise in CTEPH. 

## Figures and Tables

**Figure 1 medicina-57-00355-f001:**
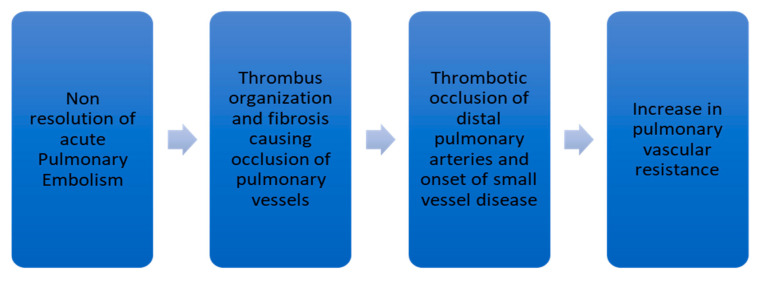
Flow diagram showing the pathophysiology of CTEPH.

**Figure 2 medicina-57-00355-f002:**
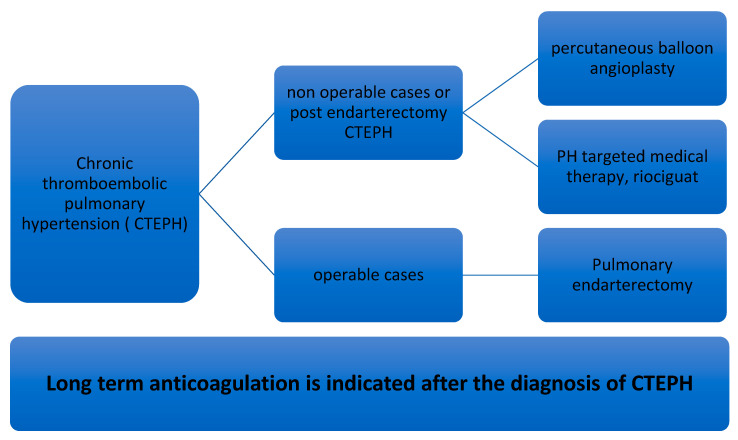
Definitive treatment options for chronic thromboembolic pulmonary hypertension.

**Figure 3 medicina-57-00355-f003:**
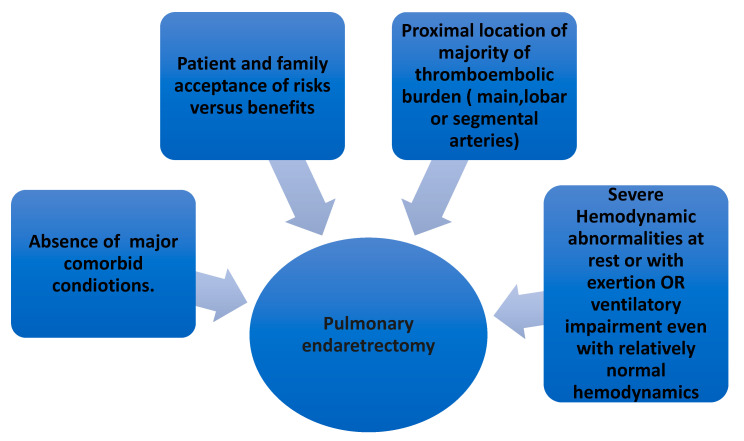
Major factors associated with the decision to proceed with pulmonary endarterectomy.

**Figure 4 medicina-57-00355-f004:**
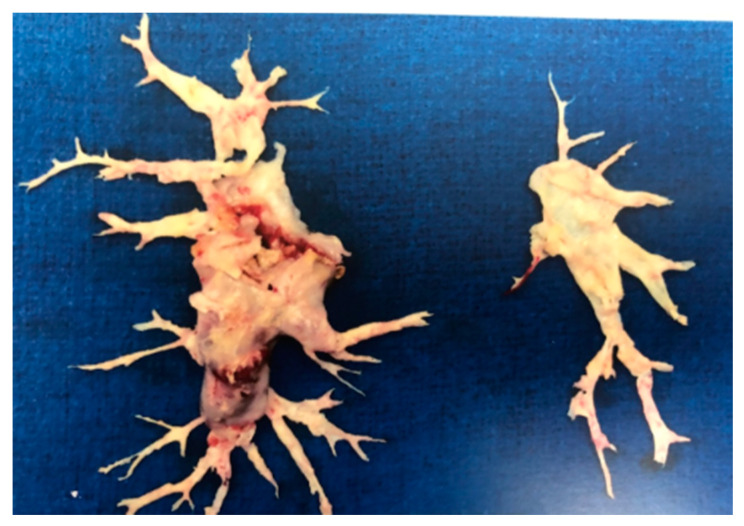
Specimen removed (both left and right) after pulmonary endarterectomy in a patient with chronic thromboembolic pulmonary hypertension.

**Table 1 medicina-57-00355-t001:** Different groups of pulmonary hypertension. Left column indicates group of pulmonary hypertension and right column shows the entity included.

Group 1	Pulmonary Arterial Hypertension (PAH)
Group 2	Due to Left Heart Disease
Group 3	Due to Chronic Lung Disease/Hypoxemia
Group 4	Due to Pulmonary Arterial Obstructions
Group 5	Multifactorial

**Table 2 medicina-57-00355-t002:** Studies estimating incidences of chronic thromboembolic pulmonary (CTEPH) after pulmonary embolism (PE).

Investigators	Year of Publication	Type of Study	Main Results
Ende-Verhaar YM et al. [3]	February 2017	Metanalysis of patients with PE followed up for CTEPH. Sample size n: 4047 patients from 16 studies	-CTEPH incidence was 3.2% (95% CI 2–4.4) of 999 patients who survived after PE ≥ 2 years (4 studies) -Pooled CTEPH incidence 0.56% (95% CI 0.1–1.0) (3 studies) -In survivors without major comorbidities, incidence was 2.8% in 1775 patients (9 studies)
Miniati M et al. [4]	September 2006	Prospective study of 320 patients with proven PE follow-up for a median duration of 2.1 years	4 out of 320 patients with proven PE (1%) developed CTEPH
Klok FA et al. [5]	January 2010	Cohort screening study in 866 patients with acute PE studied between January 2001 and July 2007	CTEPH incidence was 0.57% (95% CI, 0.02–1.2%) in all-cause PE and 1.5% (95% CI, 0.08–3.1%) in provoked PE
Pengo V. [6]	May 2004	Prospective follow-up of 223 patients with acute PE for median duration of 94.3 months. Follow-up ventilation-perfusion scan and pulmonary angiography done in patients suspected to have CTEPH	CTEPH incidence was 1% (95% CI, 0.0–2.4) at 6 months, 3.1% (95% CI, 0.7–5.5) at 1 year, and 3.8% (95 CI, 1.1–6.5) at 2 years
Berghaus TM [7]	December 2011	Cohort screening study of 43 survivors of recurrent PE	CTEPH in 5 patients (11.6%) patients with recurrent PE

**Table 3 medicina-57-00355-t003:** Some of the possible risk factors for the development of CTEPH.

Possible Risk Factors for CTEPH
(1) Large pulmonary emboli
(2) Recurrent pulmonary emboli
(3) Insufficient anticoagulation
(4) Underlying cancer
(5) Chronic inflammatory states
(6) Infections (example: staphylococcal infection)
(7) Biological and genetic risk factors
(8) Blood groups
(9) Abnormal fibrinogen levels and fibrinolysis
(10) Platelet dysfunction
(11) Platelet endothelial adhesion molecule deficiency (PECAM)

## Data Availability

Not applicable.

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
