# Peer review of "Revisiting a Distinct Entity in Pulmonary Vascular Disease: Chronic Thromboembolic Pulmonary Hypertension (CTEPH)"

_medicina, 2021, doi:10.3390/medicina57040355_

Round 1
Reviewer 1 Report
A very interesting publication. This is the review of Chronic Thromboembolic Pulmonary Hypertension (CTEPH). The current data on epidemiology, diagnosis and treatment of CTEPH are presented in an interesting way. Well-selected, up-to-date literature. The work contains several schematic, well-designed diagrams. In figure 4 there are some typographical errors. There is thrmboemolic - should be thromboembolic. There is aretries - there should be arteries. The citation location for figure 4 should be better chosen. The publication is eligible for publication after minor corrections.
Author Response
We would like to thank the reviewer for such wonderful comments on our manuscript. The typographical errors mentioned in the reviewer's comments have been corrected.
Thank you
Reviewer 2 Report
Sharma and Levine reviewed Chronic thromboembolic pulmonary hypertension in this article.
I personally do not see the gain of this Review to the literature. Entire paragraphs are referencing Reference 27. Which then turns out is duplicated to Reference 2.
The tables and Figures look very hastily put together and give me, as the reader the feeling not a lot of effort was invested.
Table 1 has no headings, and therefore unequal formatting, why is Group I in bold, whereas the rest it non-bold? There is a large gap between the table1 heading and the table itself in the document.
Table 2 is the only table in the document that should be called as such - a table - containing references and multiple columns of information.
Table 3 has a different color text, and is lacking any sort of references. Table 3 is missing a portion after 11. PECAM- as the braket doesnt close and overall I do not understand the point of a table for this portion - a text passage would be sufficient.
Figure 4 , is a single image, without much description what is left and what is right and looks like a cell-phone picture taken of a secondary image.
Author Response
Thank you for the comments by the reviewer.
CTEPH is a well known entity ( in the books) but is not commonly diagnosed in a real world clinical practice. It is often misdiagnosed or diagnosed late and that causes an immense impact in a patient's life. The whole purpose of this "review" article is to succinctly "review" and revisit this important disease entity.
Table 1 has no headings, and therefore unequal formatting, why is Group I in bold, whereas the rest it non-bold? There is a large gap between the table1 heading and the table itself in the document
(answer) The bold in group I has been corrected. The gap has been corrected. In the table description , we have added what the left part of the column in the table and right part indicate though these things are self explanatory.
Table 3 has a different color text, and is lacking any sort of references. Table 3 is missing a portion after 11. PECAM- as the braket doesnt close and overall I do not understand the point of a table for this portion - a text passage would be sufficient.
(answer) The color text has been changed to black. The bracket after PECAM has been corrected. The whole paragraph about "predisposing factors for CTEPH" is summarized in this table for convenience of the readers. There are multiple "references" in the paragraph itself for every risk factor listed !
Figure 4 , is a single image, without much description what is left and what is right and looks like a cell-phone picture taken of a secondary image.
(answer) We have added both "left and right" in the description. They both are obviously specimen removed after thrombectomy. Its well explained in the Figure legend.
Reviewer 3 Report
Diagnostic groups are denoted in Arabic numbers not Roman numerals. Needs to be corrected throughout manuscript. CTEPH is group 4 (more specifically 4.1). Table 1 is the most glaring example of the misuse. Please refer to Simonneau's publication which you cite to correct.
Please provide citations for the following statements:
Lines 50-51 "...CTEPH remains astonishingly under-reported.
Lines 70 -73 "Malignancy as risk factor" I don't think is supported by registry data or my personal experience.
Lines 201 - 202: "resting hemodynamics underestimate in CTEPH"
Instead of repetitive, "Studies show..." in paragraph lines 88 - 99, provide some comment on which risk factors have the most supportive evidence and how this might assist the clinician in the evaluation. For example, should all patients have a full hypercoagulability evaluation (not cheap!).
Consider making Table 3 on risk factors 2 columns entitled "Likely" and "Possible" to add some context for the reader.
Line 112: Should lead this paragraph with the presentation is non-specific
Lines 124-131: TR and MR murmurs are pansystolic, what distinguishes a TR murmur. Add pulmonary artery bruits.
Consider commenting on the role of BNP/NT-proBNP and D-dimer.
Lines 171-3: I don't understand the sentence comparing the sensitivity and specificity of VQ and CTA from the He study.
Lines 289: Medical therapy for CTEPH is indicated for both "inoperable" and PH that persists after PTE
Consider commenting on RACE trial for BPA.
There are numerous capitalization errors (e.g. WSPH, Titles of Tables and Figures), a few spelling errors (e.g. it is Wood units on Woods, thromboembolism in figure 3, etc.), and occasional poor sentence structure that should be corrected to make manuscript acceptable.
Author Response
We thank the reviewer for the comments.
Diagnostic groups are denoted in Arabic numbers not Roman numerals. Needs to be corrected throughout manuscript. CTEPH is group 4 (more specifically 4.1). Table 1 is the most glaring example of the misuse. Please refer to Simonneau's publication which you cite to correct.
(answer) corrected
Please provide citations for the following statements:
Lines 50-51 "...CTEPH remains astonishingly under-reported.
(answer) Incidences of CTEPH has been provided in the manuscript. The sentence referred by the reviewer has been removed.
Lines 70 -73 "Malignancy as risk factor" I don't think is supported by registry data or my personal experience.
(answer) We have cited study in ref 15. Malignancy as a risk factor has been proposed to be a possible predisposing factor by limited studies. Its not a definite statement......."Malignancy has also been suspected to be a risk factor for the development of CTEPH by limited studies. It is likely related to the increased production of cytokines, acute phase reactants, and unregulated coagulation and fibrinolytic system. In a European study that involved 687 patients, there was a strong association between development of CTEPH in patients with malignancy with the odds ratio of 3.76, 95% confidence in the wall of 1.47–10.43 [15]."
Lines 201 - 202: "resting hemodynamics underestimate in CTEPH"
(answer) This sentence has been removed. Subsequent sentences are cited ( Ref 27).
Instead of repetitive, "Studies show..." in paragraph lines 88 - 99, provide some comment on which risk factors have the most supportive evidence and how this might assist the clinician in the evaluation. For example, should all patients have a full hypercoagulability evaluation (not cheap!).
(answer) not everyone should have hyper coagulability evaluation. Thats why we have mentioned it in the manuscript.
Consider making Table 3 on risk factors 2 columns entitled "Likely" and "Possible" to add some context for the reader.
( answer) We agree with reviewer. Since even an episode of large PE doesn't necessarily mean that CTEPH will develop in the future, we have changed the entire table title to "possible" risk factors.
Line 112: Should lead this paragraph with the presentation is non-specific
(answer) In the paragraph about Clinical features we have added.."These symptoms are non-specific to CTEPH."
Lines 124-131: TR and MR murmurs are pansystolic, what distinguishes a TR murmur. Add pulmonary artery bruits.
(answer) added the location for a TR murmur ( left lower sternal border). Also added pulmonary artery bruits.
Consider commenting on the role of BNP/NT-proBNP and D-dimer.
(answer) These tests are so vague and non-specific for diagnosis of CTEPH and are implicated in so many different diseases. We doubt even mentioning these tests as diagnostic tool in a succinct "review" article would benefit the readers. Besides, we have mentioned about BNP in the medical treatment section....
Lines 171-3: I don't understand the sentence comparing the sensitivity and specificity of VQ and CTA from the He study.
(answer) Not sure what reviewer is intending to say. We have presented result of study by He et al and provided reference !!
Lines 289: Medical therapy for CTEPH is indicated for both "inoperable" and PH that persists after PTE
(answer) Ref 41
There are numerous capitalization errors (e.g. WSPH, Titles of Tables and Figures), a few spelling errors (e.g. it is Wood units on Woods, thromboembolism in figure 3, etc.), and occasional poor sentence structure that should be corrected to make manuscript acceptable.
(answer) corrected
Round 2
Reviewer 3 Report
#1 Lines 150-2: TR and MR murmurs are pansystolic, what distinguishes a TR murmur. Add pulmonary artery bruits.
(answer) added the location for a TR murmur ( left lower sternal border). Also added pulmonary artery bruits.
Please also consider adding that a typical TR murmur is enhanced with inspiration.
#2 Consider commenting on the role of BNP/NT-proBNP and D-dimer.
(answer) These tests are so vague and non-specific for diagnosis of CTEPH and are implicated in so many different diseases. We doubt even mentioning these tests as diagnostic tool in a succinct "review" article would benefit the readers. Besides, we have mentioned about BNP in the medical treatment section....
Consider mentioning that D-dimer is unhelpful (Arunthari V ... Open Respir Med J 2009).
#3 Lines 196-199: Guidelines recommend VQ for screen which Drs Sharma and Levine appropriately emphasize, but I am unconvinced that the He work is strong evidence in support of that recommendation. While sensitivity VQ 100% was quantitatively better than CTA 92.2% in the He review (2012), He and colleagues concluded that the results were “similar” and that both had excellent efficacy. By contrast, Sharma and Levine state that VQ was “better” which does not reflect the conclusion of He et al.
Author Response
#1 Lines 150-2: TR and MR murmurs are pansystolic, what distinguishes a TR murmur. Add pulmonary artery bruits.
(answer) added the location for a TR murmur ( left lower sternal border). Also added pulmonary artery bruits.
Please also consider adding that a typical TR murmur is enhanced with inspiration.
(answer) added
#2 Consider commenting on the role of BNP/NT-proBNP and D-dimer.
(answer) These tests are so vague and non-specific for diagnosis of CTEPH and are implicated in so many different diseases. We doubt even mentioning these tests as diagnostic tool in a succinct "review" article would benefit the readers. Besides, we have mentioned about BNP in the medical treatment section....
Consider mentioning that D-dimer is unhelpful (Arunthari V ... Open Respir Med J 2009).
(answer) done
3 Lines 196-199: Guidelines recommend VQ for screen which Drs Sharma and Levine appropriately emphasize, but I am unconvinced that the He work is strong evidence in support of that recommendation. While sensitivity VQ 100% was quantitatively better than CTA 92.2% in the He review (2012), He and colleagues concluded that the results were “similar” and that both had excellent efficacy. By contrast, Sharma and Levine state that VQ was “better” which does not reflect the conclusion of He et al.
(answer) corrected